# Credibility Assessment for Digital Twins in Vehicle-in-the-Loop Test Based on Information Entropy

**DOI:** 10.3390/s25051372

**Published:** 2025-02-24

**Authors:** Tianfang Gao, Liang Chen, Xinghui Zhang, Jinghua Guo, Dong Ni

**Affiliations:** 1Xiamen Product Quality Supervision and Inspection Institute, Xiamen 361023, China; gaotianfang@xmzjy.org (T.G.); chenliang@xmzjy.org (L.C.); zhangxinghui@xmzjy.org (X.Z.); 2School of Aerospace Engineering, Xiamen University, Xiamen 361005, China; guojh@xmu.edu.cn

**Keywords:** digital twins, vehicle-in-the-loop test, information entropy, approximate entropy, cross approximate entropy

## Abstract

Digital twins in vehicle-in-the-loop (VIL) test has great practical significance for the functional development, testing and evaluation of intelligent vehicle. The study about the credibility assessment of dynamically evolving models still lacks effective approaches. In addition, it has rarely been studied in automotive tests. In this paper, a closed loop test of dynamic virtual and real-world interaction was built, and its characteristics are also analyzed. According to its characteristics and assessment methods, a credibility assessment methodology based on information entropy is proposed to reveal the degree of its own information confusion and structural relevance of different information, which involves ApEn and cross-ApEn. The algorithm has been successfully verified in experiments and it has been found that the inconsistent weight of the real and digital vehicle is an important factor on digital twins VIL tests. Furthermore, the effect of the length of series on the credibility assessment has been emphatically studied, and the results show that it has no more than 2% effect on the credibility assessment.

## 1. Introduction

In 2021, Level 2 autonomous passenger vehicles have been commercialized with a penetration rate of 23.5% in China, and Level 3 or higher autonomous vehicles will enter the market in the future [1]. More and more intelligent driving technologies are being developed and studied, and various functions and performance of vehicles are also being tested and verified. Different from the traditional system, the intelligent or autonomous vehicle has become a strong coupling system of human-vehicle-environment-driving tasks. So, it has become a challenge to test these intelligent/autonomous vehicle [2].

Before an intelligent vehicle can enter the market, it must undergo a series of safety tests, as shown in Figure 1, which can be summarized as simulation tests and road tests. In simulation tests [3], Model-in-loop (MIL) and software-in-loop (SIL) tests are completely simulation, that is, sensors, vehicles, controllers and driving environment are simulated by software modeling. Hardware-in-the-loop (HIL) is often used for active safety-related controller tests. Except for the controller is real, other systems, such as vehicle dynamics, roads, driver and gearbox, are virtual, and its accuracy is low due to the deviation between the simulated and the real automotive system. The actual road test is the most accurate, but its repeatability is poor and randomness is strong. And the VIL test based on digital twins is preferred that combines simulation tests with real roads [4]. Vehicle-in-the-loop (VIL) test based on digital twins places the real vehicle in a virtual parallel world by digitally modeling the autonomous driving vehicle and scenario, which can provide an effective and economical solution for the complex whole vehicle and test scenarios. A number of implementations of VIL tests based on digital twins’ theory have been reported in the literatures [5,6,7,8,9,10,11,12].

VIL tests based on digital twins is a promising methodology for intelligent vehicle and credibility is particularly important for its application. Credibility is concerned with the confidence required by the system of digital twins VIL test to evaluate the functionality and performance of intelligent vehicle. Credibility assessment refers to the assessment process of obtaining the quantitative measurement of the confidence [13], which can shorten the research cycle of the intelligent driving system and save economic costs. Credibility assessment methods can refer to the one of simulation modeling because of their similarity. The literature [14] suggested the general framework of VV&A (Verification, Validation and Accreditation). In the literature [15], graphical comparison, confidence intervals, and hypothesis tests are the most common approaches used for model validation. The literature [16] pursued a Bayesian probabilistic approach to quantitative model validation with non-normality data, considering data uncertainty and to investigate the impact of normality assumption on validation accuracy. The methods mentioned in literature [14,15,16,17] were not appropriate for digital twins because the model remained the same during the application process. The literature [18] focused on the assessment of basic models and used Goal-Question-Metric method to established a systematic multidimensional evaluation index system for digital twin models and combined the advantages of variable fuzzy recognition model and information entropy aggregation weight algorithm to construct a digital twin quality value evaluation method based on improved variable fuzzy model. The literature [19] proposed an improved network hierarchical analysis method based on triangular fuzzy number to analyze the consistency of their physical properties. The literature [18,19] didn’t quantify the credibility according to actual applications. In book [20], MAE (Mean Absolute Error), MSE (Mean Square Error), NMSE (Normalized Mean Square Error) or RMSE (Root Mean Square Error) were used for comparing different evolution. The improved grey relational analysis (GRA) based on weighted linear [21,22] and Theil’s inequality coefficients (TIC) [23] were also used for the consideration assessment. The thesis [24] studied the model validation methods and a comprehensive method of credibility assessment [25], including the consistency analysis of model output data, similarity entropy, DS evidence theory and CWWA operator. The literature [26] proposed a method of the application-time-window (ATW) by fitting a joint probability distribution based on observation of actual application time point.

In a word, the study about the credibility assessment of dynamically evolving models still lacks effective approaches. Besides, it has rarely been studied in automotive tests. In our work, combined the essence of credibility assessment with the characteristics of VIL tests based on digital twins, a credibility evaluation method based on information entropy is proposed, and it was verified in experiments. Besides, the impact factors of the assessment algorithm were discussed and analyzed.

This article is organized as follows. Section 2 describes the construction and debug communication of VIL test based on digital twins, and the characteristics of the system are analyzed. Then in Section 3, approximate entropy (ApEn) and cross approximate entropy (cross-ApEn) is introduced, and credibility algorithm is described in detail. After that, a laboratory experimental test was devised to verify the effectiveness of the methodology in Section 4, containing Scenario A and Scenario B. Furthermore, the impact factors (especially the length of the series N) of the assessment algorithm were discussed. Finally, Section 5 concludes with a brief summary and discussion.

## 2. System Construction and Feature Analysis

To quantitatively evaluate the credibility of VIL test based on digital twins, the system is firstly introduced, mainly including real vehicle, closed proving ground, simulation platform and communication port. The construction is also detailed. Then, its characteristics are analyzed, involving operations among digital twin layer, real vehicle layer and user layer.

### 2.1. System Construction of VIL Test Based on Digital Twins

The system of VIL test based on digital twins that focuses on accurately reproducing closed proving ground tests of intelligent vehicles was built as shown in Figure 2 and Figure 3. The VIL test system based on digital twins includes real vehicle, closed proving ground, simulation platform and communication port. Real vehicle is the intelligent vehicle equipped with GNSS/INS integrated navigation system, which integrates global navigation satellite system and inertial navigation system and achieves high-precision attitude information of the vehicle using Kalman filter. The digital vehicles continuously receive the information about the attitude and position through this integrated navigation system. Closed proving ground can support vehicle road testing, such as roads, traffic signs, traffic obstacles, traffic lights and etc. The high-definition map of the closed proving ground was taken to establish a one-to-one digital twins test platform, involving the road and its ancillary facilities. It can be guaranteed that the coordinate system of closed proving ground and simulation platform is consistent. Simulation platform consists of mapping module, road module, environment module, vehicles & sensors module, traffic participants module and acquisition/reception module. The intelligent driving system of real vehicle receives information from GNSS/INS integrated navigation system and sends the vehicle information to the simulation platform through network cable. Mapping module continuously collects the latitude &longitude and attitude information of the real vehicle movement, and converts them into Cartesian coordinates based on the Albers position mapping method, mapping the real world to the virtual environment. Acquisition/reception module is completed through the communication port, which can not only export the simulated test scene data, but also accept the data of the braking, steering and others fed back by the GNSS/INS.

This test system can form a closed loop of dynamic virtual and real world interaction through injecting the test scenarios generated by the virtual traffic environment (including virtual objects, traffic, and environment) into the real vehicle controller and mapping motion behavior generated by real vehicle to the virtual traffic environment. The construction of the VIL test system based digital twins is established through the following steps, as shown in Figure 4.

Design a functional model of the test vehicle, as shown in Figure 5. The real vehicle is mapped into the virtual scenario by means of the digital twin technology to ensure that real vehicle and virtual vehicle have the same dynamic characteristics. The di-mensions of the vehicle are: 5.0 m × 1.5 m × 2.2 m (L × W × H). The real vehicle sends motion results to the digital one every 10 ms through Ethernet, CAN or wireless network to ensure the consistency of them;Design virtual scenarios for testing. Road map can be obtained by creating, importing or editing the high-precision map and also using surveying and mapping data creation or virtual editing. A variety of sensor parameters and traffic signs are added according to the requirements, and the target information detected by the virtual sensors is injected into the real vehicle controller in the form of CAN bus;Debug communication. The synchronization of the test system is independently completed by three main parts: the mapping system, the simulation software and the target injection system. The cycle time of three processes, including mapping the position and attitude information of the real vehicle to simulation software, simulation software outputting environment and target information, should be less than the vehicle message cycle. Through debugging, the message cycle was set to 10 ms to synchronize software and hardware, and also to decrease data dropout;Start testing, save and analyze test data.

### 2.2. Feature Analysis of VIL Test Based on Digital Twins

Data updates and information exchanges is a distinctive feature of digital twin VIL test system, and the flow chart is shown in Figure 6. In the figure, the digital twin test system mainly includes three parts: digital twin layer, real vehicle layer and user layer. Digital vehicle is being updated in digital twin layer by sensing data to realize the dynamic mapping of real vehicle. At the same time, the environment and behavior obtained by simulation platform should also react to real vehicle, sending directive information to complete the prediction and control of real vehicle. Digital twin layer updates the parameters of original digital model, and the updated model is used to predict the performance, continuously updating data and advancing the testing process.

The data-driven characteristics of digital twin VIL test system not only require the accurate acquisition, generation, analysis and transmission of virtual information and physical information, but also emphasize the consistency and real-time nature of the information exchanges. Information exchanges make real-time and dynamic digital model true. Real vehicle test layer needs to dynamically transmit information such as structural input and current running status to digital twin layer in real time while digital twin layer needs to accurately and timely transmit information such as the results of fault diagnosis, evaluation and prediction and behavior control to real vehicle test layer.

In conclusion, the system of VIL test based on digital twins is one that combines virtual and physical systems based on information exchanges driven by data updates. The credibility of VIL system based on digital twins can be evaluated by information, including quantity, degree of confusion and structure, which can be obtained through the consistency of information entropy between practical and digital twin VIL test system.

## 3. Credibility Assessment Methodology

The essence of credibility assessment of VIL test based on digital twin is to how to measure information, including content and randomness. Entropy is the fundamental concept in information theory, which quantifies the amount of information and measures the degree of randomness in the system. ApEn is not intended to be an approximation of the entropy but a statistic for the measure of regularity whose foundations are similar and correctly quantifies finite data series; it was devised as a quantification of the rate of regularity in time data series [27]. Approximate entropy (ApEn) has been used widely in medicine, finance, mechanical systems and others [28].

### 3.1. Information Entropy

#### 3.1.1. Approximate Entropy

Approximate entropy (ApEn) is an indicator of signal complexity [29], which is suitable for extracting features in nonlinear systems, and is widely used in signal processing, analysis, computer science and so on [30,31]. It is a statistic, and the greater the signal complexity, the greater the ApEn.

For time series signals $\left\{x\left(i\right),i=1,2\ldots ,N\right\}$, the ApEn is calculated as following. *m* is the number of dimensions, *r* is the compatibility limit. An *m*-dimensional vector $X\left(i\right)$ is constructed from the original time series:(1)$$X(i)=\left[x\left(i\right),x\left(i+1\right),\ldots ,x\left(i+m-1\right)\right]\left(i=1,2,\ldots ,N-m+1\right)$$

For each *i*, the distance *d* is calculated between $X\left(i\right)$ and $X\left(j\right)\left(j=1,2,\ldots ,N-m+1\right)$, and the number of $d\left[X\left(i\right),X\left(j\right)\right]>r$ is counted.(2)$$d\left[X\left(i\right),X\left(j\right)\right]=max\left\|X_{i+k}-X_{j+k}\right\|,k=0,1,2\ldots ,m$$(3)$$C_{i,m,r}=\frac{1}{N-m+1}num\left\{d\left[X\left(i\right),X\left(j\right)\right]>r\right\}\;\left(j=1,2,\ldots N-m+1\right)$$(4)$$\varphi^{m}\left(r\right)=\frac{\sum_{i=1}^{N-m+1}C_{i,m,r}}{N-m+1}$$(5)$$C_{AE}=\varphi^{m}\left(r\right)-\varphi^{m+1}\left(r\right)$$

The dimension is increased to *m* + 1 and repeat Equations (1)–(4) to obtain the value of $\varphi^{m+1}\left(r\right)$, and the ApEn is shown in Equation (5). Generally, *m* is generally 2, *r* is taken as (20~25%) of the standard deviation of original time series. And in such case, the value of $C_{AE}$ has the least dependence on the length of the series *N*.

#### 3.1.2. Cross Approximate Entropy

Cross approximate entropy (cross-ApEn) is used for distinguishing the degree of similarity and applicable to condition monitoring and fault diagnosis of mechanical equipment [32,33]. It can quantitatively describe the cross-correlation degree between irregularity and complexity of time series signals, and better reflect the dynamic and intrinsic characteristics of the signals. The smaller the cross-ApEn is, the more similar the data, which means that the two sets of data are about closer.

There are two time series vectors $\;\left\{x_{i}\right\}$ and $\left\{y_{i}\right\}$, and multidimensional vectors is constructed as follows:(6)$$X\left(i\right)=\left(x_{i},x_{i+1},\ldots ,x_{i+m-1}\right),i=1,2,\ldots ,N-m-1$$(7)$$Y\left(j\right)=\left(y_{j},y_{j+1},\ldots ,y_{j+m-1}\right),j=1,2,\ldots ,N-m-1$$

The distance between these two vectors is defined:(8)$$d_{ij}=max\left\|X_{i+k}-Y_{j+k}\right\|,k=0,1,2\ldots ,m$$

A method for calculating the compatibility limit r is given:(9)$$r=0.2*COV\left(X_{i},Y_{j}\right)$$

For each *j*, $C_{i,m,r}$ is counted to obtain the ratio of vectors with a distance less than *r*, and $\varphi^{m}\left(r\right)$ can be calculated by Equation (4). The correlation between the two sets of data can also be calculated by Equation (5).

### 3.2. Assessment Algorithm Based on Information Entropy

Traditionally, the credibility evaluation of digital twin models relies on the consistency of the output data between practical test and digital twins VIL test, including structure and function. Because the characteristics of digital twin VIL test system are data-driven and real-time update, it should be analyzed in terms of the amount, confusion and structure of the information in the perspective of informatics.

The credibility evaluation can be verified by combining the complexity and correlation of information obtained from the practical and digital twins VIL test. Considering the quantity, structure and complexity, the information entropy of digital twin VIL test system is the difference between the ApEn of all output data and cross-ApEn between any two variables. $T_{EVS}$ is information entropy of VIL test system based on digital twins, $T_{EPS}$ is information entropy of practical VIL test system. The credibility evaluation of VIL test based on digital twins can be calculated from Equations (10)–(12), as shown in Figure 7.(10)$$T_{EVS}=\sum_{i=j}C_{AE}\left(X_{i},X_{j}\right)-\sum_{i\neq j}C_{AE}\left(X_{i},X_{j}\right)$$(11)$$T_{EPS}=\sum_{i=j}C_{AE}\left(Y_{i},Y_{j}\right)-\sum_{i\neq j}C_{AE}\left(Y_{i},Y_{j}\right)$$(12)$$I_{CR}=1-\left|\frac{T_{EVS}-T_{EPS}}{T_{EPS}}\right|\times 100\%$$

### 3.3. Assessment Process for Digital Twins in VIL Test

The credibility assessment process of digital twin models is illustrated as following:Analyze the type and quantity of data output by VIL test system, and filter the non-abnormal data. The data acquisition equipment is required to work normally.Calculate the ApEn and cross-ApEn of digital twin and practical VIL test system, and get the mapping correlation of the dataset.Evaluate the credibility of the model. When calculating information entropy, be careful not to miss any items.Accurately evaluate the credibility of VIL test system based on digital twins. A large number of repeated experiments are allowed for a comprehensive and objective evaluation. For the same VIL test based on digital twins, there are often differences in the information entropy due to the randomness and uncertainty of the output data of practical VIL test and VIL test based on digital twins. The mean and mean square deviation of credibility can be used to evaluate VIL test based on digital twins.

## 4. Experiments and Discussion

Automatic Emergency Braking System (AEB) is an active safety system that can automatically detect the risk of collision and autonomously brake to avoid collision. It is an important part of autonomous vehicles below Level 3 and is currently the most widely used for driving assistance system. At present, domestic and international standards/regulations of AEB test are mostly oriented to straight road test scenarios. The specific control strategy of AEB test is as follows: when the actual relative distance is less than the preset safety distance, the AEB system applies automatic emergency braking. The evaluation indicators include relative distance of the target, the collision time TTC, and braking deceleration.

The practical VIL test and VIL test based on digital twins are validated in Figure 8. In Scenario A and Scenario B, the road surface is a good, dry asphalt concrete one, and surface adhesion coefficient is not less than 0.9. The initial speed of the test vehicle is 0, the constant driving speed rate is 5 m/s, the acceleration is 1 m/s^2^. The initial distance between test vehicle and target obstacle (person or vehicle) is 100 m, the target obstacle is always stationary, and the test vehicle approaches the stationary obstacle at a constant speed of 5 m/s. The safety distance is 8.5 m. The data acquisition system of GNSS/INS integrated navigation system records data every 10 ms, including velocity of test vehicle and relative distance between test and target vehicle. In order to reduce the impact of randomness and uncertainty of digital twin/actual system on the credibility assessment, three groups of digital twins VIL and practical road tests are carried out respectively.

### 4.1. Results of Scenario A

Figure 9a is the speed of test vehicle for both test systems, and Figure 9b is the relative distance between test vehicles and target vehicle for both test systems. Theoretically, the vehicle will brake before 18.3 s. In the Figure 9b, both test vehicles activate the AEB system and the test vehicle of digital twin VIL test brakes earlier than practical test. The curves of the speed and relative distance have a good consistency for both test systems. The collision distance of digital twin VIL system is about 11 m while the one of practical test system is about 5 m. The AEB algorithm of digital twin vehicle is based on the data at the lowest level, and directly outputs the target-level results after fusion perception, which is very accurate.

According to the proposed method, the ApEn and cross-ApEn for both systems are calculated separately, as shown in Table 1, and the corresponding credibility evaluation is shown in Table 2. The information entropy fluctuation of the practical road test is greater than that of the digital twins VIL test. Finally, the reliability of digital twin VIL system is 75.68%, and the mean square deviation is 4.00%. And for the same practical road test, there is little difference in credibility assessment. However, the credibility of digital twin VIL test model is relatively low. It was founded to be caused by the inconsistent weight of the real and digital vehicle, which can be verified in Scenario B.

### 4.2. Results of Scenario B

After Scenario A, it is noted that the weight consistency between the actual and real vehicles care should be taken care of, especially the number of people in the vehicle. For Scenario B, Figure 10a is the speed of test vehicle for both test systems, and Figure 10b is the relative distance between test vehicles and target vehicle for both test systems. Theoretically, the vehicle will brake before 18.3 s. It can be seen that the braking time of the digital twin VIL test is similar to the actual road test, and the collision distance is also similar. The collision distance of digital twin VIL system is about 9.8 m while the one of practical test system is about 9.2 m. In Figure 10b, AEB of the test vehicle in digital twin VIL test performed slightly better than real road test.

According to the proposed method, the EnAp and cross-ApEn for both systems are calculated separately, as shown in Table 3, and the corresponding credibility evaluation is shown in Table 4. The information entropy fluctuation of digital twins VIL and practical road test is no more than 0.1. Finally, the reliability of digital twin VIL system is 96.12%, and the mean square deviation is 2.87%. After adjustment, the credibility of the model has increased, which also indirectly proves the reliability of the algorithm. Whether the obstacle is a dummy or a vehicle, the recognition algorithm of vehicle is based on the software’s own algorithm, and there is no difference.

### 4.3. Discussion of Impact Factors

The reliability of the credibility quantification algorithm can be seen in Section 4.1 and Section 4.2, and the impact factors are discussed and analyzed in this subsection. As can be seen from Section 3, the factors that affect the results of credibility assessment include the length of the series N. the number of dimensions m and the compatibility limit r. The compatibility limit r is a threshold, which will affect the statistics that meet the conditions and in turn has influence on assessment results. For the calculation of ApEn, *r* is taken as 20% of the standard deviation [34]; For the calculation of cross-ApEn, *r* is taken as 20% of the covariance of the compared sequences [32,33]; the number of dimensions *m* is generally 2 [35].

To discuss the impact of the length of the series *N* on information entropy and credibility assessment, the first set of Digital twins VIL and practical road test data (Scenario B) were taken for an example. Considering the completeness of the testing process, the length of the series *N* starts from 2500 to 3000 with an interval of 100. Figure 11 is entropy of different length *N* in digital twins VIL test, Figure 12 is entropy of different length N in practical road test. As seen in Figure 11 and Figure 12, As the length increases, the ApEn of the velocity increases by about 0.015 while the ApEn of the relative distance decreases by about 0.005. And the cross-ApEn of υ&d decreases by about 0.015 with the increasement of the data length. As a result, the value of credibility fluctuates by no more than 2%.

## 5. Conclusions

A VIL test based on digital twins was introduced, which injects the test scenarios generated by the virtual traffic environment into the real vehicle controller and mapping motion behavior generated by real vehicle to the virtual traffic environment. The real and digital vehicles continuously have the same attitude and position through GNSS/INS integrated navigation system. The system of VIL test based on digital twins is updated by data, and its distinctive feature are data updates and information exchanges. So that, the credibility of VIL system based on digital twins can be evaluated by information, including quantity, degree of confusion and structure, which can be obtained through the consistency of information entropy between practical and digital twins VIL test system.

In order to accurately assess the credibility of digital twins VIL test, a method based on information entropy is proposed in terms of quantity, degree of confusion and structure for information exchanges. The credibility algorithm involves ApEn and cross-ApEn, revealing respectively the degree of its own information confusion and structural relevance of different information. With the proposed algorithm, the credibility of VIL test based on digital twins in Scenario A is 75.68% while one in Scenario B is 96.12%. It has been founded that the inconsistent weight of the real and digital vehicle is an important factor on VIL test based on digital twins, especially the number of people in the vehicle.

The factors that affect the results of credibility assessment have also be studied, including the length of the series N, the number of dimensions *m* and the compatibility limit *r*. The influence of the length of series N on the credibility assessment was emphatically analyzed, and the result show that it has no more than 2% effect on the credibility assessment.

In the future, other parameters of VIL test based on digital twins will be analyzed on the influence of credibility assessment, which in turn facilitates the accuracy of VIL test based on digital twins. An object classification approach of information entropy can also be considered for autonomous vehicles in machine learning techniques.

## Figures and Tables

**Figure 1 sensors-25-01372-f001:**
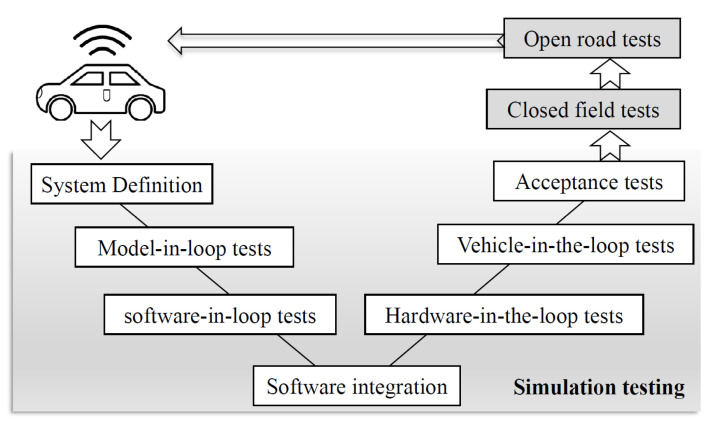
V-shaped process for autonomous driving system development.

**Figure 2 sensors-25-01372-f002:**
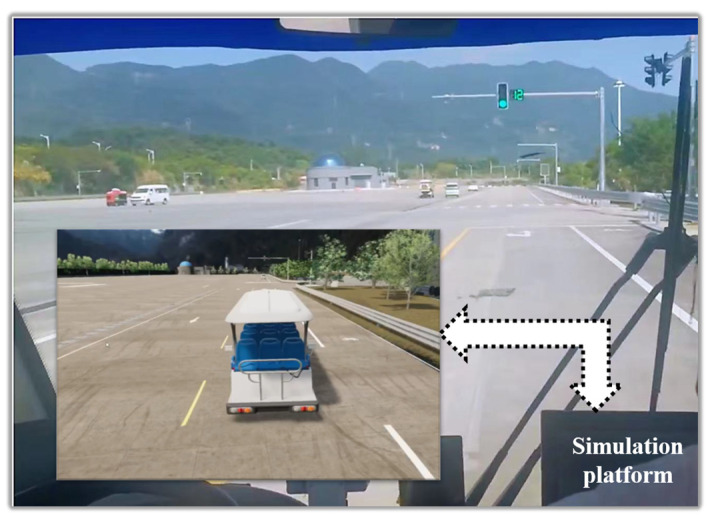
The VIL test based on digital twins.

**Figure 3 sensors-25-01372-f003:**
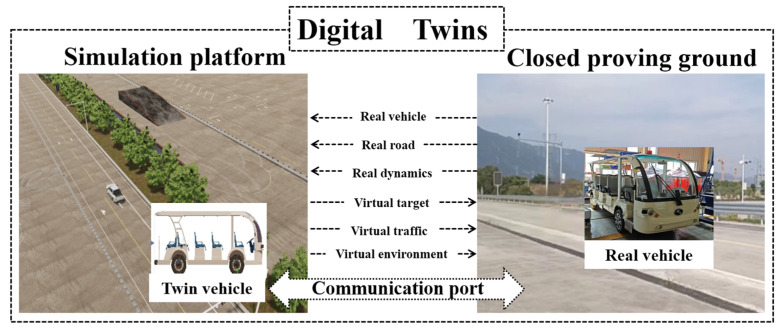
The system framework of VIL test based on digital twins.

**Figure 4 sensors-25-01372-f004:**
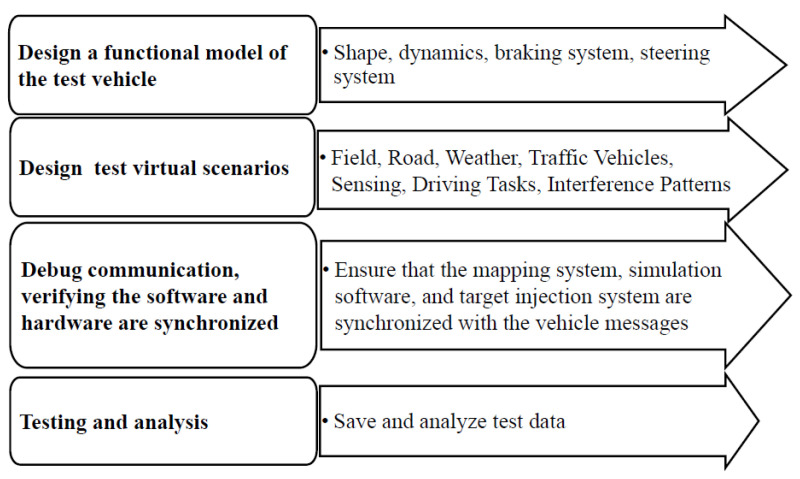
System construction of VIL test based on digital twins.

**Figure 5 sensors-25-01372-f005:**
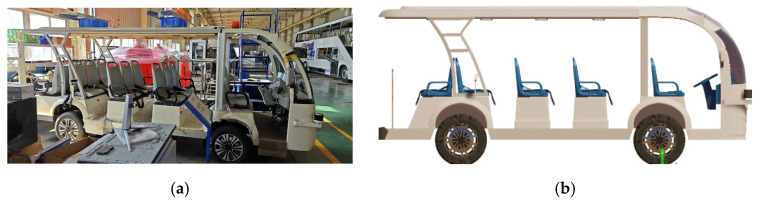
(**a**) Test vehicle; (**b**) Digital test vehicle.

**Figure 6 sensors-25-01372-f006:**
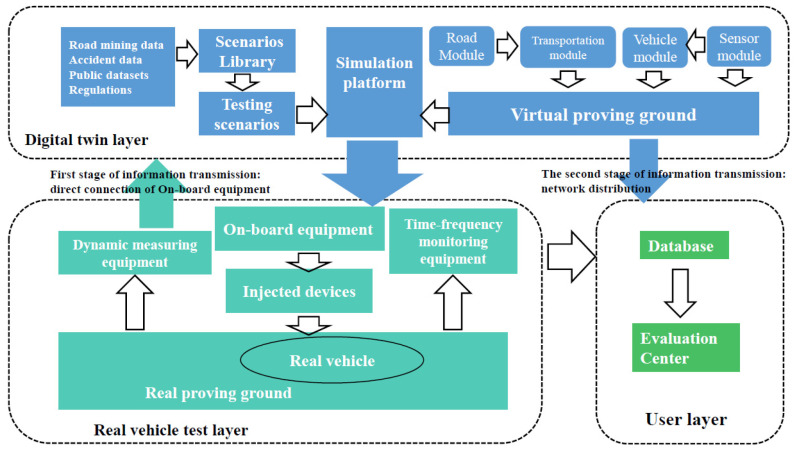
Information data flow of Digital Twin VIL test.

**Figure 7 sensors-25-01372-f007:**
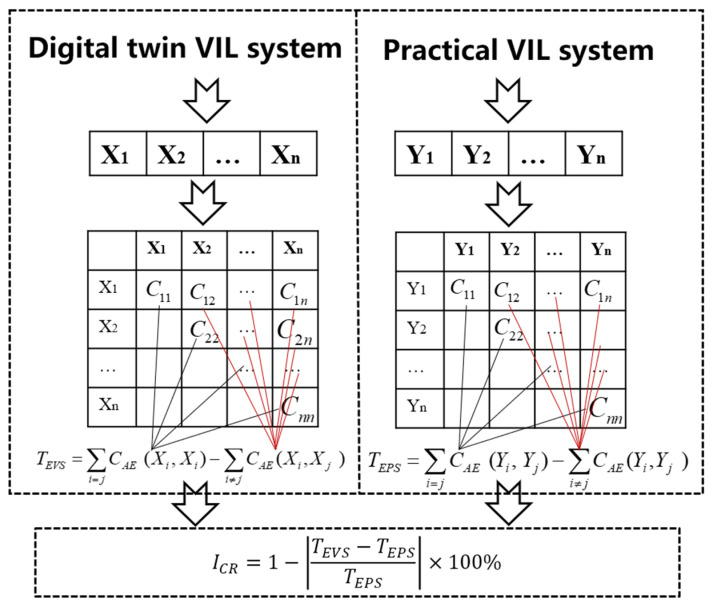
The credibility evaluation algorithm.

**Figure 8 sensors-25-01372-f008:**
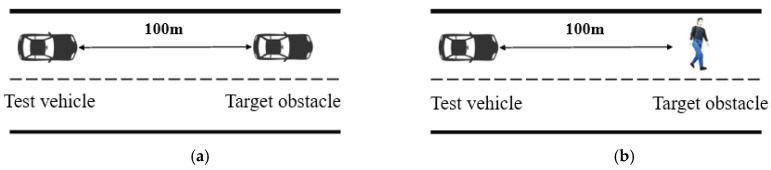
(**a**) Scenario A: stationary obstacle is vehicle; (**b**) Scenario B: stationary obstacle is dummy.

**Figure 9 sensors-25-01372-f009:**
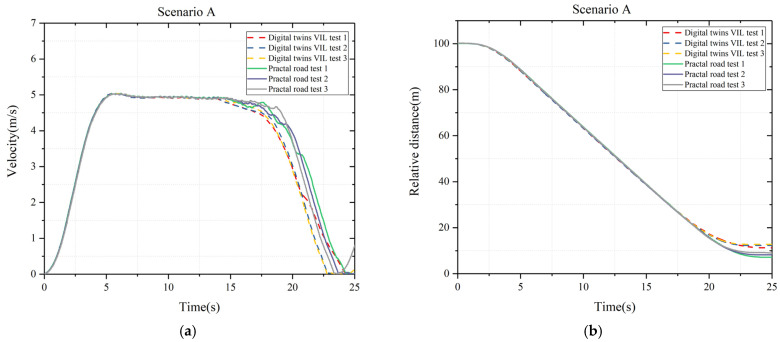
Scenario A: (**a**) velocity of test vehicle; (**b**) relative distance between test and target vehicle.

**Figure 10 sensors-25-01372-f010:**
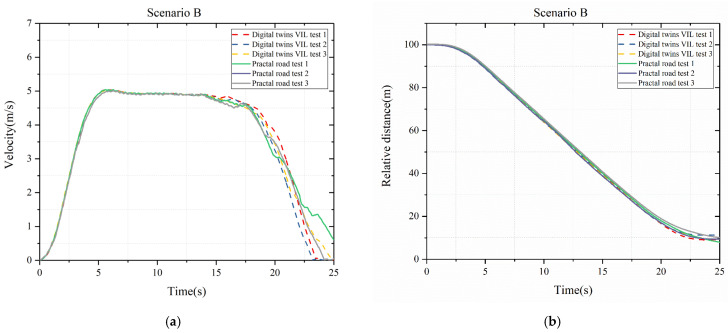
Scenario B: (**a**) Velocity of test vehicle; (**b**) Relative distance between test and target vehicle.

**Figure 11 sensors-25-01372-f011:**
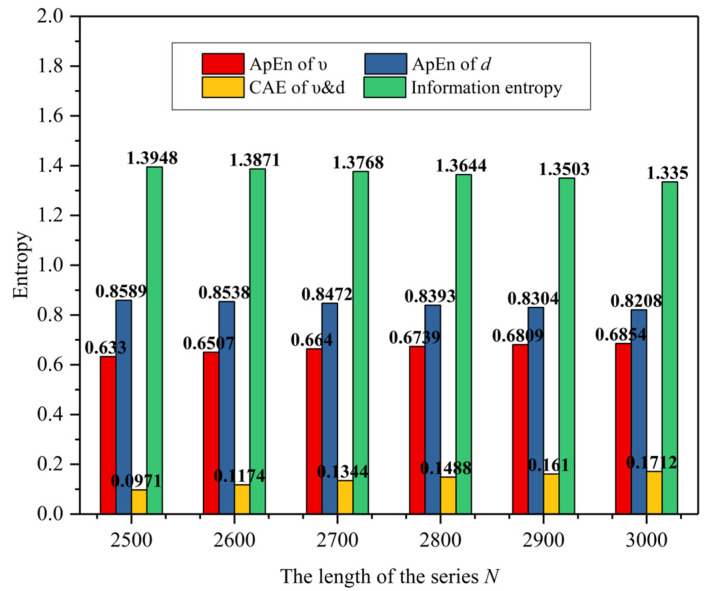
Entropy of different length N in digital twins VIL test.

**Figure 12 sensors-25-01372-f012:**
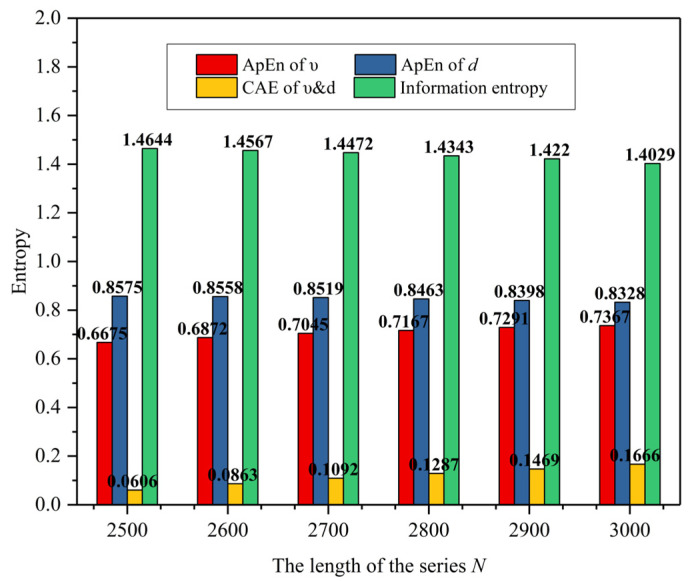
Entropy of different length N in practical road test.

**Table 1 sensors-25-01372-t001:** ApEn, cross-ApEn and information entropy of Scenario A.

Scenario A		ApEn		cross-ApEn	Information Entropy
		υ	*d*	υ&d	
Digital twins VIL system	1	0.7270	0.8047	0.0807	1.4510
	2	0.7190	0.8014	0.0528	1.4676
	3	0.7241	0.8012	0.0583	1.4670
Actual road test system	1	0.6930	0.7888	0.3534	1.1284
	2	0.7315	0.7896	0.3163	1.2048
	3	0.7171	0.7918	0.3113	1.1976

**Table 2 sensors-25-01372-t002:** Credibility evaluation of Scenario A.

Credibility Evaluation Scenario A		Practical VIL Test		
		1	2	3
Digital twin VIL system	1	71.41%	79.57%	78.84%
	2	69.94%	78.18%	77.45%
	3	69.99%	78.23%	77.51%

**Table 3 sensors-25-01372-t003:** ApEn, cross-ApEn and information entropy of Scenario B.

Scenario B		ApEn		Cross-ApEn	Information Entropy
		υ	*d*	υ&d	
Digital twins VIL system	1	0.6878	0.8108	0.1797	1.3189
	2	0.7097	0.8056	0.0899	1.4254
	3	0.7219	0.8114	0.1595	1.3738
Actual road test system	1	0.7434	0.8182	0.1875	1.3741
	2	0.7621	0.8251	0.1828	1.4044
	3	0.7559	0.8195	0.1184	1.457

**Table 4 sensors-25-01372-t004:** Crebibility evaluation of Scenario B.

Credibility Evaluation Scenario B		Practical VIL Test		
		1	2	3
Digital twin VIL system	1	95.98%	93.91%	90.52%
	2	96.27%	98.50%	97.83%
	3	99.97%	97.82%	94.29%

## Data Availability

Data are contained within the article.

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
