# Peer review of "Credibility Assessment for Digital Twins in Vehicle-in-the-Loop Test Based on Information Entropy"

_sensors, 2025, doi:10.3390/s25051372_

Round 1
Reviewer 1 Report
Comments and Suggestions for Authors
In this paper, a closed loop test of dynamic virtual and real-world interaction was built, and its characteristics are also analyzed. the effect of the length of series N on the credibility assessment has been emphatically studied, and the results show that it has no more than 2% effect on the credibility assessment.
1. The contribution should be be set in a separate paragraph.
2. How the vehicle platform connect to the digital-twin platform? How to decrease the time-delay and data dropout?
3. How is the coordinate system guaranteed to be consistent, between the vehicle platform and digital-twin platform?
4. The connection animations between the vehicle platform and digital-twin platformshould be showed.
5.In Fig 10, the velocity seems can not be followed when it brakes, how to get the vehicle velocity should be explained(CAN bus or GNSS), and the prediction algorithm should also be added to ensure the velovity can be followed by digital-twin platform.
Author Response
Manuscript ID: sensors-3439415
Type of manuscript: Article
Title: A Credibility Assessment Methodology of Digital-twin Technology-based
Vehicle-in-the-loop Test based on Information Entropy
Authors: Tianfang Gao, Liang Chen, Xinghui Zhang, Jinghua Guo, Dong Ni *
Dear Editor,
Thank you for managing our manuscript (Manuscript ID: sensors-3439415). We are grateful to Reviewers for their insightful comments. They have considerably contributed to the quality of the manuscript, which has been thoroughly revised. All corrections are marked in red. The point-by-point answers to the comments and suggestions have been listed as below:
Reviewer 1
In this paper, a closed loop test of dynamic virtual and real-world interaction was built, and its characteristics are also analyzed. the effect of the length of series N on the credibility assessment has been emphatically studied, and the results show that it has no more than 2% effect on the credibility assessment.
Question 1: The contribution should be set in a separate paragraph.
Response: Thanks for the kindly suggestion of the reviewer. The contribution have been set in a separate paragraph at Lines 345-370.
Question 2: How the vehicle platform connect to the digital-twin platform? How to decrease the time-delay and data dropout?
Response: Thanks for careful comments of the reviewer. The intelligent driving system of real vehicle is connected to the digital-twin platform by network cable at Lines 116-118. The time-delay and data dropout can be achieved by debug communication at Lines 149-155.
Question 3: How is the coordinate system guaranteed to be consistent, between the vehicle platform and digital-twin platform?
Response: Thanks for kindly suggestion of the reviewer. The coordinate system of closed proving ground and simulation platform is consistent by taking the high-definition map of the closed proving ground to establish a one-to-one digital twins test platform. We have added it at Lines 111-114.
Question 4: The connection animations between the vehicle platform and digital-twin platform should be showed.
Response: Thanks for kindly suggestion of the reviewer. The connection animations between the vehicle platform and digital-twin platform have been added in Figure 2.
Question 5: In Fig 10, the velocity seems can not be followed when it brakes, how to get the vehicle velocity should be explained(CAN bus or GNSS), and the prediction algorithm should also be added to ensure the velocity can be followed by digital twin platform.
Response: Thanks for careful comments of the reviewer. In Fig.10, it is because of the instability of the intelligent driving system that the velocity seems can not be followed when it brakes. The velocity of test vehicle was achieved by GNSS/INS integrated navigation system at Lines 270-272. In the system of VIL test based on digital twin, the real and digital vehicles continuously have the same attitude and position through GNSS/INS integrated navigation system of real vehicle at Lines 116-118. As long as the debugging is successful (at Lines 149-155), the velocity of real and digital vehicle can be considered synchronized.

Reviewer 2 Report
Comments and Suggestions for Authors
The paper introduces and experimentally analyzes a closed-loop test of a dynamic virtual and real-world interaction. The tested experiments and their environment have been clearly identified and presented in the paper.
Some modifications are required to be improved:
1- The title may need to be reconsidered.
2- A separate section is required regarding the previous studies that have considered this field of research needs to be introduced (Related Work). Besides some applications of intelligent vehicles on the road consider:
-
Detecting urban road condition and disseminating traffic information by VANETs
D-Hop: A dynamic and distributed protocol for vehicle routing
A performance evaluation of a context-aware path recommendation protocol for Vehicular Ad-hoc Networks
An Object Classification Approach for Autonomous Vehicles Using Machine Learning Techniques
A distributed infrastructure-based congestion avoidance protocol for Vehicular Ad Hoc Networks
2- Section 2 and Section 3 need to have more informative titles. Besides before starting the subsections, a short introduction paragraph is required.
3- The benefits of the proposed work and its usability need to be clearly discussed.
4- The references should follow the format and some more references need to be considered in this field.
Comments on the Quality of English Language
The English needs significant proofreading. e.g., "And" should not start a sentence, etc.
Author Response
Manuscript ID: sensors-3439415
Type of manuscript: Article
Title: A Credibility Assessment Methodology of Digital-twin Technology-based
Vehicle-in-the-loop Test based on Information Entropy
Authors: Tianfang Gao, Liang Chen, Xinghui Zhang, Jinghua Guo, Dong Ni *
Dear Editor,
Thank you for managing our manuscript (Manuscript ID: sensors-3439415). We are grateful to Reviewers for their insightful comments. They have considerably contributed to the quality of the manuscript, which has been thoroughly revised. All corrections are marked in red. The point-by-point answers to the comments and suggestions have been listed as below:
Reviewer 2
The paper introduces and experimentally analyzes a closed-loop test of a dynamic virtual and real-world interaction. The tested experiments and their environment have been clearly identified and presented in the paper. Some modifications are required to be improved:
Question 1: The title may need to be reconsidered.
Response: Thanks for the kindly suggestion of the reviewer. We have changed the title to “Credibility Assessment for Digital twins in Vehicle-in-the-loop Test based on Information Entropy”.
Question 2: A separate section is required regarding the previous studies that have considered this field of research needs to be introduced (Related Work). Besides some applications of intelligent vehicles on the road consider:
- Detecting urban road condition and disseminating traffic information by VANETs
- D-Hop: A dynamic and distributed protocol for vehicle routing
III. A performance evaluation of a context-aware path recommendation protocol for Vehicular Ad-hoc Networks
- An Object Classification Approach for Autonomous Vehicles Using Machine Learning Techniques
- A distributed infrastructure-based congestion avoidance protocol for
Vehicular Ad Hoc Networks
Response: Thanks for the kindly suggestion of the reviewer. We have modified it at Lines 50-86. Reviewers' suggestions are considered at Lines 369-370.
Question 3: Section 2 and Section 3 need to have more informative titles. Besides before starting the subsections, a short introduction paragraph is required.
Response: Thanks for the kindly suggestion of the reviewer. We have modified it at Lines 95,101,157,182,191,219 and 238. A short introduction of Section 2 is at Lines 96-100. A short introduction of Section 3 is at Lines 183-190.
Question 4: The benefits of the proposed work and its usability need to be clearly discussed.
Response: Thanks for the suggestion of the reviewer. We have explained it at Lines 183-190.
Question 5: The references should follow the format and some more references need to be considered in this field.
Response: Thanks for the careful suggestion of the reviewer. We have modified it at Lines 379-456.
Comments on the Quality of English Language
The English needs significant proofreading. e.g., "And" should not start a sentence, etc.
Response: Thanks for the careful suggestion of the reviewer. We have modified it.

Round 2
Reviewer 1 Report
Comments and Suggestions for Authors
All comments from previous review rounds were reflected in the revised manuscript. Some references relevant to this research could also be included.
LIU JX, GAO B L, ZHONG W, et al. Adaptive optimization strategy and evaluation of vehicle-road collaborative perception algorithm in real-time settings[J], Computers and Electrical Engineering, 2024, 120:109785.
LU Y B, LIANG J H, ZHUANG W C, et al. Four-wheel independent drive vehicle fault tolerant strategy using stochastic model predictive control with model parameter uncertainties[J]. IEEE Transactions on Vehicular Technology, 2024, 73(3):3287-3299 .